# Impact of N-Linked Glycosylation on Therapeutic Proteins

**DOI:** 10.3390/molecules27248859

**Published:** 2022-12-13

**Authors:** Baoquan Chen, Wenqiang Liu, Yaohao Li, Bo Ma, Shiying Shang, Zhongping Tan

**Affiliations:** 1State Key Laboratory of Bioactive Substance and Function of Natural Medicines, Institute of Materia Medica, Chinese Academy of Medical Sciences and Peking Union Medical College, Beijing 100050, China; 2Center of Pharmaceutical Technology, School of Pharmaceutical Sciences, Tsinghua University, Beijing 100084, China

**Keywords:** glycosylation, proteins, erythropoietin, monoclonal antibodies

## Abstract

Therapeutic proteins have unique advantages over small-molecule drugs in the treatment of various diseases, such as higher target specificity, stronger pharmacological efficacy and relatively low side effects. These advantages make them increasingly valued in drug development and clinical practice. However, although highly valued, the intrinsic limitations in their physical, chemical and pharmacological properties often restrict their wider applications. As one of the most important post-translational modifications, glycosylation has been shown to exert positive effects on many properties of proteins, including molecular stability, and pharmacodynamic and pharmacokinetic characteristics. Glycoengineering, which involves changing the glycosylation patterns of proteins, is therefore expected to be an effective means of overcoming the problems of therapeutic proteins. In this review, we summarize recent efforts and advances in the glycoengineering of erythropoietin and IgG monoclonal antibodies, with the goals of illustrating the importance of this strategy in improving the performance of therapeutic proteins and providing a brief overview of how glycoengineering is applied to protein-based drugs.

## 1. Introduction

According to the central dogma, the genetic information carried on DNA is transcribed to RNA and then translated to protein, with protein generally considered the functional end product in the process. However, it has been demonstrated that this process is actually far more complex than what is initially described by the central dogma. The number of expressed proteins could be orders of magnitude greater than the number of protein-coding genes, due to alternative splicings, variable promoter usage, post-translational modifications (PTMs), and other regulatory mechanisms. Among these mechanisms, PTMs are important contributors to the vast diversity of proteomes and can lead to an exponential increase in the complexity of the proteome, relative to that of the transcriptome or genome. A wide range of PTMs, including phosphorylation, glycosylation, ubiquitination, acetylation, and methylation, have been identified [1] (Figure 1). Of these, glycosylation is the most common and complex PTM on secreted proteins. It is estimated that about 85% of secreted proteins are glycosylated [2].

Protein glycosylation occurs mainly in the endoplasmic reticulum (ER) and the Golgi apparatus, where glycosyl donors are covalently linked to target glycosyl acceptors (such as proteins and lipids) through enzyme-catalyzed processes involving approximately 200 glycosyltransferases [3]. There are two main types of protein glycosylation: N-linked glycosylation (N-glycosylation) and O-linked glycosylation (O-glycosylation). In N-glycosylation, the glycans are covalently attached to the side-chain nitrogen (N) atoms of the Asn residues in the N-X-S/T sequons, where X is any amino acid except proline. In O-glycosylation, the side-chain oxygen (O) atoms of the Ser/Thr residues are used as the connection points for the glycans.

Unlike the synthesis of DNA, RNA and protein, glycosylation is not a template-driven process. It is regulated by many different factors, such as the relative accessibility of potential glycosylation sites and the availability of activated glycosyl donors and glycosyltransferases [4]. Due to the lack of tight control of the process of glycan biosynthesis, glycoproteins secreted from cells usually exist as complex mixtures of up to a hundred different glycosylated protein isoforms (glycoforms), which differ in their glycosylation sites and/or glycan structures [5,6]. The compositions of glycoform mixtures can vary significantly depending on the cell types and expression conditions [5,6]. Different glycoforms have different biological properties and functions [7,8]. It thus appears quite possible to develop new therapeutic proteins, or to improve the efficacy of existing protein-based drugs, by changing the glycosylation patterns of proteins (glycoengineering).

Therapeutic proteins as macromolecules have favorable characteristics, such as higher specificity, better efficacy and lower side effects, compared to the small-molecule drugs that have been used clinically for centuries. This is why they are now widely accepted and administered to patients with cancer or other life-threatening diseases [9]. However, due to their complex structures and large sizes, therapeutic proteins also have unfavorable characteristics, such as limited solubility, stability [10] and biological properties, which could lead to less desirable outcomes in clinical use, and ineffective or even harmful treatments. Substantial efforts have been devoted to minimizing these problems [11]. With a continuously increasing number of tools available for manipulating glycosylation sites and glycan structures (glycosylation patterns), glycoengineering has become an attractive strategy for achieving such a goal [12,13].

Previous glycoengineering efforts have demonstrated the feasibility of this strategy [14]. However, the number of successful applications has so far been limited, due to insufficient understanding of the structure–function relationship of protein glycosylation, and a lack of reliable scientific theories to guide the glycoengineering design process. To date, the most well-known examples of glycoengineering are erythropoietin (EPO) and immunoglobulin G (IgG) antibodies. In this review, we will summarize and discuss current knowledge about the glycoengineering of these two types of representative therapeutic proteins, with the goals of providing a brief overview of the studies undertaken and the current status of this research area, and of facilitating the future application of glycoengineering to develop more successful protein-based drugs. In addition to these two representative examples, there are many coagulation factors, cytokines, and hormone-based therapeutic proteins whose properties have been reported to be affected by glycosylation. A detailed description of these reported findings is beyond the scope of this mini review, and the interested reader is referred to the excellent recent review articles for more comprehensive information [14,15,16,17,18].

## 2. Erythropoietin

Human erythropoietin (HuEPO) is a cytokine. It is mainly secreted by renal interstitial cells, but a small amount can also be synthesized by hepatocytes (Figure 2A). Its expression is regulated by the blood oxygen level and controlled by the hypoxia-inducible transcription factor-1 (HIF-1) [19,20]. HuEPO was first isolated and purified by Goldwasser et al. in 1977 from the urine of patients with aplastic anemia [21]. Subsequent studies found that the HuEPO gene encodes a protein precursor of 193 amino acids. Cleavage of a 27-amino-acid signal peptide from the N-terminus of this precursor yields a protein of 166 amino acids [22] (Figure 2B). The C-terminal arginine residue is proteolytically removed prior to secretion, resulting in a mature protein of 165 amino acids.

HuEPO consists of four α-helices and contains two disulfide bonds, one between Cys7 and Cys161, and the other between Cys29 and Cys33. Natural HuEPO is a heavily glycosylated protein. It has a much higher molecular weight (about 30 kDa) than the unglycosylated one (about 18 kDa), with the glycan moiety comprising approximately 40% of its total molecular weight. HuEPO has three N-glycosylation sites at Asn24, Asn38 and Asn83, and one O-glycosylation site at Ser126 [23]. Characterization of the glycosylation of HuEPO revealed that the three N-glycans on this protein typically have highly sialylated bi-, tri- or tetra-antennary structures, and the O-glycan has a mucin-core-1-type structure. The same as most other glycoproteins, HuEPO isolated from human urine always exists as heterogeneous mixtures of glycoforms. Different glycoforms have been demonstrated to have different properties and functions. For example, it was found that HuEPO glycoforms treated with sialidase, which catalyzes the removal of the terminal sialic acid residues from glycoproteins, had no in vivo erythropoietic activity [24].

It was not possible to obtain sufficient amounts of HuEPO from human urine to meet the clinical needs of patients. In order to address this gap, in 1985, Jacobs and Lin successfully cloned and expressed the gene encoding HuEPO [25,26]. This achievement made it possible to produce recombinant human erythropoietin (rHuEPO) in Chinese hamster ovary cells on a manufacturing scale. In 1989, rHuEPO was approved by the US Food and Drug Administration (FDA) for clinical use under the trade names Epogen^®^/Procrit^®^ and Eprex^®^. Since then, rHuEPO has become one of the most successful glycoprotein drugs. It is now widely employed for the treatment of anemia of various causes, such as renal anemia and tumor-related anemia. This application has changed the way of treating patients with end-stage renal disease on chronic hemodialysis, where blood transfusion treatment used to be the only means of survival. The administration of rHuEPO not only improves hemoglobin levels and anemia symptoms, but also strongly stimulates bone marrow erythroid progenitor cells to increase the number of mature red blood cells [27] (Figure 2A).

As a typical therapeutic protein, rHuEPO shares the same disadvantages as other protein-based drugs. For example, it must be administered by injection and, because of a relatively short half-life, multiple injections are required to maintain an effective therapeutic level, which frequently leads to low quality of life for patients [28]. One of the main parameters responsible for the short half-life of EPO is believed to be the rate of body clearance. At present, the exact pathway via which EPO is removed from the circulation has not been fully elucidated. It is generally speculated that this mainly occurs in the liver and kidneys, and is mediated by receptors, such as the EPO receptor (EPOR), on the cell surface. Previous findings suggest that the interaction between EPO and EPOR promotes the cellular uptake and degradation of rHuEPO through endocytosis, and that the disappearance rate of rHuEPO is directly related to the number of EPORs: if the total number of EPORs in chemotherapy patients is low, the EPO clearance rate is also low [29]. Studies have also shown that EPO glycoforms lacking sialic acid could be recognized and rapidly cleared by asialoglycoprotein receptors (ASGPR) on the surface of hepatocytes [30]. However, more than 90% of glycans in rHuEPO are fully sialylated, so the ASGPR-mediated process may be the major mechanism for the clearance of rHuEPO, only after sialic acid is removed from the serum by sialidase [31].

To improve the compliance of anemia patients, novel EPO derivatives with extended in vivo half-lives have long been the focus of research in the field of medicine. Inspired by the observation that the molecular size of glycans, and the number of sialic acids, have a significant effect on the clearance rate of proteins, a glycoengineered long-lasting EPO derivative, darbepoetin alfa (trade name Aranesp^®^), was developed and launched by Amgen in 2001. Compared to natural human EPO, darbepoetin alfa has two additional N-linked glycans at positions 30 and 88 (Figure 2B) [32]. This change increases its half-life in the circulation by a factor of 3, and reduces the frequency of its administration to once every 1 or 2 weeks [33]. The improvement in the properties of darbepoetin alfa makes it a very successful therapeutic agent. In 2021, the global sales of Aranesp^®^ reached USD 1.5 billion [34].

The enhanced and prolonged biological effects of darbepoetin alfa are apparently due to a greater resistance to degradation and not due to a higher binding affinity for EPOR. Its binding affinity to EPOR is actually lower than that of rHuEPO, which is likely to be the result of the increased glycan density and sialic acid content. The large size of the glycans, together with their dynamic properties, may have the capability of sterically hindering the interactions between darbepoetin alfa and EPOR. At the same time, the increased charges carried by the additional sialic acid residues may also have a negative effect on the binding to EPOR [35]. Overall, the combined effects of these two factors may act to partially reduce EPOR-binding-mediated endocytosis, and thus increase the half-life of darbepoetin alfa.

The additional two sialylated N-linked glycans at Asn30 and Asn88 increase the size of darbepoetin alfa. This increase is believed to contribute to the improvement in the pharmacokinetics of the drug, which is likely to be related to the renal clearance of proteins (which occurs primarily through glomerular filtration). The glomerular filtration rate decreases as the size of the protein increases, and the molecular-weight threshold limiting the glomerular filtration is approximately 40 Å [36]. When the size of a protein is small, it can readily pass through the glomerular filtration barrier under normal conditions, exhibiting a rapid clearance from the circulation. When the size approaches 40 kDa, the glomerular filtration rate drops significantly. The introduction of two additional N-glycans changes the molecular weight of darbepoetin alfa by approximately 10 kDa, to about 40 kDa, which is a 22% increase compared to rHuEPO. At the same time, the N-glycans can occupy a large space, further increasing the overall size of darbepoetin alfa. Its larger size is apparently effective in reducing the glomerular filtration rate. The circulation time of darbepoetin alfa is extended from 8 h to about 25 h [37].

Glycoengineering as a strategy to increase protein size with a view to reducing the glomerular filtration rate also has some limitations. First of all, if the protein is too small, the addition of a large amount of glycans is required to make it possible to extend the size of the protein to the level approaching the glomerular filtration threshold. Such modification could significantly affect the interaction between the protein and its targets, by mechanisms such as the steric blocking of binding sites. Second, the effect of the added glycans on the glomerular filtration rate depends on many factors, including the glycosylation site and the glycan orientation. In order to achieve an optimal effect, the properties and functions of a series of glycoforms, with many different glycosylation patterns, should be analyzed and compared. The preparation of these glycoforms could be time-consuming and costly. Third, if the protein size exceeds 40 kDa, further increasing its size by glycoengineering is unlikely to contribute much to the prolongation of the circulation time.

Protein size change alone is not sufficient to fully explain the significantly prolonged clearance time of darbepoetin alfa. Studies also suggest that the clearance time may be closely related to the level of sialylation [38]. The terminal sialic acid residues of circulating glycoproteins can protect them against clearance by ASGPR, thus leading to longer serum half-lives. Darbepoetin alfa contains 5 N-linked glycans and up to 22 sialic acids. In addition to increasing the size of this therapeutic protein, the highly sialylated glycans also contribute to suppressing the binding of ASGPR, thereby inhibiting the endocytosis mediated by ASGPR and subsequent degradation by lysosomal proteases. Another type of receptor that is involved in the elimination of glycoproteins is the mannose receptor (ManR) [39]. It can recognize glycans with mannose as their terminal residues. Again, terminal sialylation can minimize the binding of ManR and prolong the action of proteins.

## 3. Monoclonal Antibodies

rhuEPO is a relatively small protein, with a molecular weight of about 30 kDa. Its biological activity can be increased by introducing additional N-linked glycosylation sites onto the protein surface to prolong its circulation in the blood. However, such a glycoengineering strategy is not equally useful for large therapeutic proteins, such as monoclonal antibodies (mAbs), which have an average molecular weight of approximately 150 kDa. Therefore, the glycoengineering studies in the area of mAbs are not focused on extending the circulation time by introducing new glycans onto their surface, but rather on fine-tuning the structures of glycans that are naturally found on mAbs. Accordingly, the primary task in the glycoengineering of therapeutic antibodies is to gain a better understanding of the correlation between glycan structure and function.

It is well known that the immune system consists of a variety of cells, organs, and pro- and anti-inflammatory mediators throughout the body. These components form complex networks that interact with and modulate each other through cascades and positive and negative feedback mechanisms to maintain normal inflammation and immunity. Exogenous or endogenous stresses may disrupt this delicate balance, leading to the development of various immunological diseases. Traditionally, these diseases are treated by the administration of non-specific immunosuppressive and immunomodulatory agents, such as glucocorticoids, to regulate immune response. Although effective, such treatment may induce side effects due to the non-specific nature of the agents. In order to prevent the side effects and reach the desired treatment results, in recent years, more specific immunomodulatory therapies, such as mAbs, have been developed [40].

Antibodies play a central role in the function of the human immune system. They can bind to a variety of soluble antigens and block the antigens from binding to receptors on human cells (Figure 3A). They are also able to induce malignant or infected cell death through complement-dependent cytotoxicity (CDC), antibody-dependent cellular cytotoxicity (ADCC), and phagocytosis [41,42]. There are five different classes of antibodies that have been identified in humans: immunoglobulins G (IgG), IgM, IgA, IgE, and IgD. They share the same basic four-chain structure, but have different heavy chains (Figure 3B). IgG has the functions of recognizing, neutralizing and eliminating threats, and is the most abundant immunoglobulin in human serum. It accounts for about 75% of the total human immunoglobulin, and most therapeutic antibodies are of the IgG class. Adalimumab (trade name Humira^®^), the world’s first fully human therapeutic mAb, is based on the IgG1 isotype. It was approved by the FDA in 2002 for the treatment of rheumatoid arthritis (RA), and its sales reached USD 22 billion in 2021 [43].

IgG antibodies are composed of two light chains and two heavy chains, which are arranged into two Fab (fragment antigen binding) regions and one Fc (fragment crystallizable) region (Figure 3B). The Fc region is mainly responsible for interacting with various receptors and complement proteins [44]. It is composed of the second and third constant domains of the heavy chains (C_H_2 and C_H_3). The Fc region of IgG bears a highly conserved N-glycosylation site at Asn297, within the C_H_2 domain, that is essential for Fc-receptor-mediated activity [45]. The same as is observed in most glycoproteins; glycosylation at Asn297 is also highly heterogeneous, with more than 30 different glycan structures detected in the serum IgG. The glycans that are covalently linked to the Asn297 residue contain a common heptaglycan biantennary core structure (G0, four GlcNAc and three mannose residues). The core structure can be further extended differently by fucosylation (G0F), galactosylation of one or two arms (G1, G2), and addition of terminal sialic acids in the presence of galactose (G1S1, G2S2) (Figure 3C). The extended structures can differ greatly in their percentages. For example, in the consistency evaluation of 381 batches of recombinant adalimumab manufactured by AbbVie from 2000 to 2013, it was found that the terminally ungalactosylated N-glycans (G0F + G0F-GlcNAc), terminally galactosylated N-glycans (G1F + G2F), and terminally mannosylated N-glycans (M5 + M6) accounted for 74.28% ± 1.75, 18.45% ± 1.80 and 7.29% ± 0.76 of the total glycans attached to the Asn297 residue, respectively [46].

It is generally believed that the diverse N-glycan structures could confer different biological effects to therapeutic antibodies. These effects may be beneficial for the treatment of diseases by improving the therapeutic properties, or may adversely affect the biological functions [47]. For example, it was found that altered IgG glycosylation patterns in mice and humans were often accompanied by autoimmune diseases, such as rheumatoid arthritis, especially when the structures of the glycans lack the terminal sialic acid and galactose residues (G0). At the same time, it is also known that intravenous immunoglobulin (IVIG), a purified IgG fraction obtained from healthy donors, has anti-inflammatory properties, and that high-dose IVIG can be used for the treatment of autoimmune neutropenia in childhood and autoimmune hemolytic anemia [48]. In 2006, Ravetch and coworkers showed that the distinct properties (pro-inflammatory versus anti-inflammatory properties) observed for IgG antibodies are likely to be the result of the differential sialylation of the N-linked glycan at Asn297 in the Fc domain [49]. Conformational studies revealed that glycosylation may be essential for the binding of IgG Fc to Fcγ receptors by stabilizing the conformation of the heavy chains [50,51]. In addition to altering the pro- and anti-inflammatory activities of antibodies, the glycosylation of IgG Fc at Asn297 also has a profound influence on ADCC, which could be triggered by the binding of the Fc domain to the receptor FcγRIIIa. In a study assessing the effect of fucosylation on the properties of the Rituximab biosimilar Truxima, the results showed that the binding affinity of Fc to FcγRIIIa, and the ADCC activity, tend to negatively correlate with the level of core fucosylation [52].

IgG can also be glycosylated in two Fab regions. The Fab is composed of two constant domains (C_H_1 and C_L_), as well as variable heavy (V_H_) and light (V_L_) domains. About 15–25% of the IgG antibodies in human serum are N-glycosylated in the variable domains [53]. Similar to the structures of glycans present on the Fc region, the majority of the N-linked glycans found on the Fab regions are also of the complex biantennary type. The most striking difference between the glycosylation of the Fc and Fab domains lies in the extension of the core heptaglycan. The percentages of bisecting GlcNAc and terminal galactose and sialic acid are higher in Fab glycans, while the percentages of core fucose are lower. Overall, the Fab glycans are more complex and heterogeneous than the Fc glycans.

Results from previous studies have provided initial evidence that N-glycans can influence many properties of Fab. For example, it was found that the N-linked glycosylation, introduced by somatic hypermutation (SHM) in the V_H_/V_L_ regions of the autoantibodies isolated from patients with rheumatoid arthritis, could modulate the binding of Fab to the antigen citrullinated histone (cit-H2B) [54]. In another example, the antigen binding was tested using several anti-adalimumab and anti-infliximab antibody mutants, in which the naturally occurring Fab glycans were removed. The results showed that although some Fab N-glycans have no measurable effect on antigen binding, the presence of some Fab glycans, especially those in anti-adalimumab antibodies, could lead to higher binding affinity to their antigens [55]. Different mechanisms may account for the effect of Fab glycosylation on their binding affinity to different antigens, including the steric hindrance effect caused by the bulky glycans and the charge–charge interaction caused by the terminal sialic acids. 

Fab N-glycans can also play a role in increasing the stability of antibodies. In a study comparing the differences between the thermostability of anti-adalimumab and anti-infliximab antibodies with naturally acquired Fab glycans and mutants without Fab glycans, three out of five tested mutants showed lower melting temperatures [56]. Studies have also suggested Fab N-glycosylation may affect the aggregation propensity, solubility and in vivo half-life of mAbs [57,58]. However, these conclusions are presently based on a limited body of evidence and further research is needed to define the effects of these glycans.

## 4. Conclusions

As one of the most widely occurring and complex post-translational modifications, glycosylation has recently attracted great attention, especially in the field of therapeutic proteins, because of its capability to simultaneously improve multiple properties [59,60]. Theoretically, it is possible to maximize the performance of therapeutic proteins by optimizing their glycosylation patterns (glycosylation sites and glycan structures) through glycoengineering. The validity of this hypothesis has been demonstrated by the development of Darbepoetin alfa and IgG. However, similar successful examples are very rare, especially in the area of therapeutic antibody discovery and development. This situation is mainly due to the lack of deep knowledge of the structure function of antibody glycosylation, and the lack of reliable and simple glycoengineering tools. In order to address these problems, more research efforts should be devoted to gaining a better understanding of antibody glycosylation and to continuing to develop protein glycoengineering technology.

## Figures and Tables

**Figure 1 molecules-27-08859-f001:**
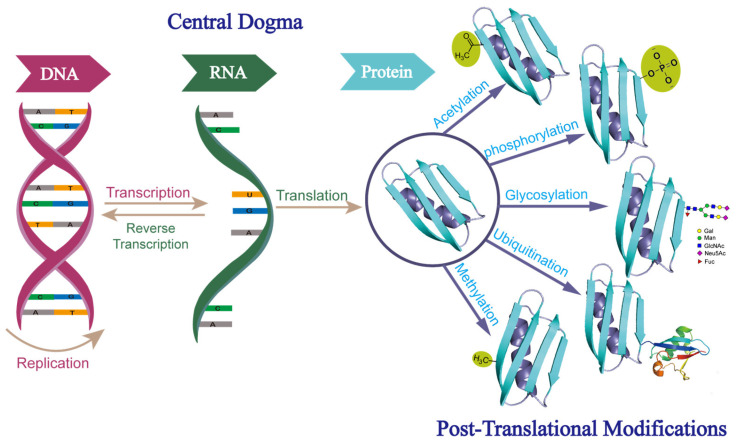
Central dogma of molecular biology and different forms of post-translational modifications. Abbreviations: Gal, galactose; Man, mannose; GlcNAc, N-acetylglucosamine; Neu5Ac, sialic acid; Fuc, fucose.

**Figure 2 molecules-27-08859-f002:**
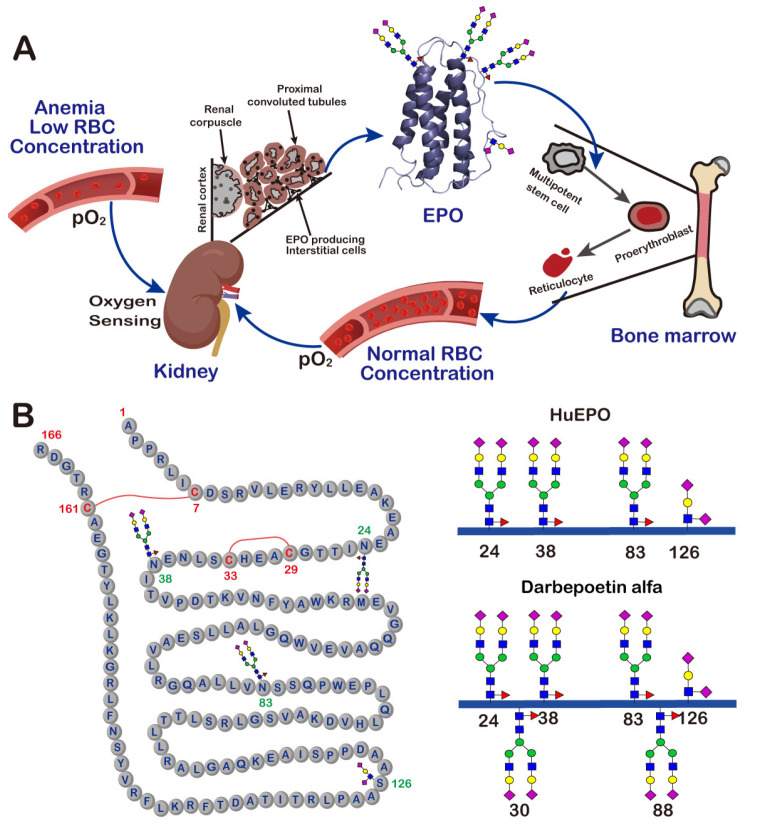
The production, function and structure of EPO. (**A**) A schematic view of the feedback loop mechanism of EPO production and function. (**B**) The amino acid sequence and glycan structures of different EPO variants.

**Figure 3 molecules-27-08859-f003:**
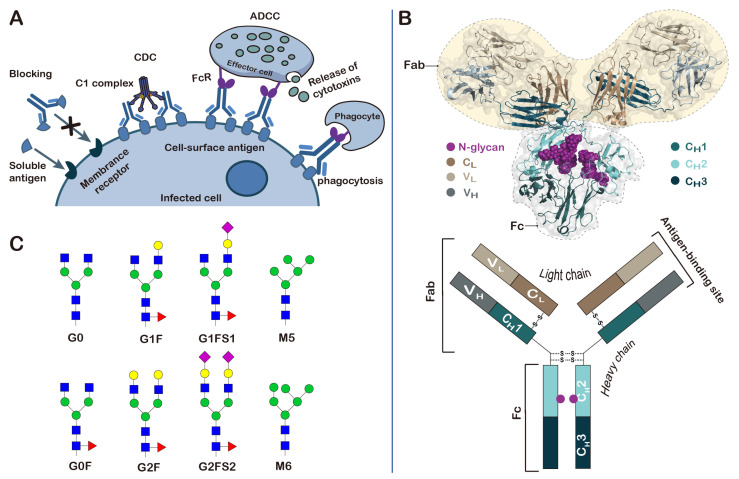
Structure, function and glycosylation of antibodies. (**A**) Schematic representation of the biological functions of antibodies; (**B**) a representative three-dimensional structure of IgG (PDB ID: 1IGY); (**C**) the N-glycan structures found on the IgG antibodies.

## Data Availability

Not applicable.

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
