# Peer review of "Impact of N-Linked Glycosylation on Therapeutic Proteins"

_molecules, 2022, doi:10.3390/molecules27248859_

Round 1
Reviewer 1 Report
This review provides a nice overview of N-glycosylation of therapeutic proteins and antibodies. I found the review easy to read and informative. The authors provide sufficient background material for readers to getup to speed and also to understand where matters stand in the therapeutic protein world. Although I realize that the review was not exhaustive, it still only focused on EPO and Humira. Concepts were brought up, but only as they pertained to these two proteins (in which I include Abs). I would have found it helpful to have a third section which includes non-approved glycoproteins and antibodies which would point the direction for the future. For example, are there any glycoproteins that have been shown to be affected by changes in glycosylation, which are not treated here? There are many examples and I suggest that the authors include some examples which are not commercial ones because they help expand this picture. A similar section could be included for Ab's.
In addition, I think the authors missed some references from the Prestegard and Barb groups showing that glycan motion/flexibility can be correlated with Ab binding and therefore hints at protein activity/function. These references can be found below.
11. Nature Chemical Biology 2011 Vol. 7 Issue 3 Pages 147-153
22. Structure 2014 Vol. 22 Issue 10 Pages 1478-1488
Author Response
This review provides a nice overview of N-glycosylation of therapeutic proteins and antibodies. I found the review easy to read and informative. The authors provide sufficient background material for readers to getup to speed and also to understand where matters stand in the therapeutic protein world. Although I realize that the review was not exhaustive, it still only focused on EPO and Humira. Concepts were brought up, but only as they pertained to these two proteins (in which I include Abs). I would have found it helpful to have a third section which includes non-approved glycoproteins and antibodies which would point the direction for the future. For example, are there any glycoproteins that have been shown to be affected by changes in glycosylation, which are not treated here? There are many examples and I suggest that the authors include some examples which are not commercial ones because they help expand this picture. A similar section could be included for Ab's.
We thank the reviewer for the valuable comments and suggestions. We totally agree that this review will be more comprehensive if the suggested information is included. However, due to the length limit of a minireview, we are not able to include so much additional information in the manuscript. We have added the following sentences to address the comments and suggestions (shown in italic):
“...In this review, we will summarize and discuss current knowledge about the glycoengineering of these two types of representative therapeutic proteins, with the goal of providing a brief overview of the studies undertaken and the current status of this research area, and to facilitate future application of glycoengineering to develop more successful protein-based drugs. In addition to these two representative examples, there are many coagulation factor, cytokine, hormone based therapeutic proteins whose properties have also been reported to be affected by glycosylation. A detailed description of these findings is beyond the scope of this minireview, and the interested reader is referred to the excellent recent review articles for more comprehensive information [14-18].”
In addition, I think the authors missed some references from the Prestegard and Barb groups showing that glycan motion/flexibility can be correlated with Ab binding and therefore hints at protein activity/function. These references can be found below.
- Nature Chemical Biology 2011 Vol. 7 Issue 3 Pages 147-153
- Structure 2014 Vol. 22 Issue 10 Pages 1478-1488
We thank the reviewer for this helpful comment. We have added these two references to the manuscript:
“...Conformational studies revealed that the glycosylation may be essential for the binding of IgG Fc to Fcγ receptors by stabilizing the conformation of the heavy chains [49,50]...”
Reviewer 2 Report
In this review, Tan and co-workers summarize the roles of N-glycans on therapeutic proteins. As exemplified by two important types of proteins, EPO and monoclonal antibodies, the authors point out that glycoengineering is a valuable strategy that could be applied in the design and evaluation of protein-based drugs. Overall, the manuscript is nicely organized and well written. Before the publication of this paper, there are a few minor issues suggested to be revised.
1. Lines 141-143: the authors propose that “the ASGPR-mediated process may not be the major mechanism of the clearance of glycosylated rHuEPO”. In fact, sialidase activity has been detected in human serum (doi: 10.3390/biology9080184). It is possible that the sialic acids on rHuEPO could be removed in serum after its administration, then ASGPR can recognize the lactose terminated EPO and resulted in clearance in liver. Unless further evidence is revealed, it is unreasonable to make such a claim.
2. Line 22, change “Antibodies” to “antibody”, or change “proteins” to “Proteins”, which should be consistent.
3. Figure 1, in the cartoon showing O-phosphorylation, maybe an oxygen atom (O) could be added between the peptide backbone and the atom P to avoid possible confusions.
4. Lines 50-54, the statement “Due to the lack of…the cell types and expression conditions” needs proper reference(s).
Author Response
In this review, Tan and co-workers summarize the roles of N-glycans on therapeutic proteins. As exemplified by two important types of proteins, EPO and monoclonal antibodies, the authors point out that glycoengineering is a valuable strategy that could be applied in the design and evaluation of protein-based drugs. Overall, the manuscript is nicely organized and well written. Before the publication of this paper, there are a few minor issues suggested to be revised.
Lines 141-143: the authors propose that “the ASGPR-mediated process may not be the major mechanism of the clearance of glycosylated rHuEPO”. In fact, sialidase activity has been detected in human serum (doi: 10.3390/biology9080184). It is possible that the sialic acids on rHuEPO could be removed in serum after its administration, then ASGPR can recognize the lactose terminated EPO and resulted in clearance in liver. Unless further evidence is revealed, it is unreasonable to make such a claim.
We thank the reviewer for this helpful comment. We have changed the following sentence to address this comment (shown in italic):
“...However, more than 90% of glycans in rHuEPO are fully sialylated, so the ASGPR-mediated process may be the major mechanism of the clearance of rHuEPO after sialic acid is removed in the serum by sialidase [30].”
Line 22, change “Antibodies” to “antibody”, or change “proteins” to “Proteins”, which should be consistent.
We have changed “proteins” in Line 22 to “Proteins”.
Figure 1, in the cartoon showing O-phosphorylation, maybe an oxygen atom (O) could be added between the peptide backbone and the atom P to avoid possible confusions.
We have added an O atom to the phosphate group in Figure 1.
Lines 50-54, the statement “Due to the lack of…the cell types and expression conditions” needs proper reference(s).
We have added the following references to the statement.
- Rudd, P.M.; Dwek, R.A. Glycosylation: heterogeneity and the 3D structure of proteins. Crit Rev Biochem Mol Biol 1997, 32, 1-100, doi:10.3109/10409239709085144..
- Goh, J.B.; Ng, S.K. Impact of host cell line choice on glycan profile. Crit Rev Biotechnol 2018, 38, 851-867, doi:10.1080/07388551.2017.1416577..